# Hollow fiber-combined glucose-responsive gel technology as an in vivo electronics-free insulin delivery system

Akira Matsumoto [1,2,11✉], Hirohito Kuwata[3,4,11], Shinichiro Kimura[3,5,11], Hiroko Matsumoto[1,2], Kozue Ochi[3], Yuki Moro-oka[1], Akiko Watanabe[3], Hironori Yamada[1], Hitoshi Ishii[6], Taiki Miyazawa [1], Siyuan Chen [1,2], Toshiaki Baba[7], Hiroshi Yoshida[7], Taichi Nakamura[8], Hiroshi Inoue[9], Yoshihiro Ogawa[3,5], Miyako Tanaka [3,10], Yuji Miyahara[1] & Takayoshi Suganami [3,10✉]

Accumulating evidence demonstrates that not only sustained elevation of blood glucose levels but also the glucose fluctuation represents key determinants for diabetic complications and mortality. Current closed-loop insulin therapy option is limited to the use of electronics-based systems, although it poses some technical issues with high cost. Here we demonstrate an electronics-free, synthetic boronate gel-based insulin-diffusion-control device technology that can cope with glucose fluctuations and potentially address the electronics-derived issues. The gel was combined with hemodialysis hollow fibers and scaled suitable for rats, serving as a subcutaneously implantable, insulin-diffusion-active site in a manner dependent on the subcutaneous glucose. Continuous glucose monitoring tests revealed that our device not only normalizes average glucose level of rats, but also markedly ameliorates the fluctuations over timescale of a day without inducing hypoglycemia. With inherent stability, diffusion-dependent scalability, and week-long & acute glucose-responsiveness, our technology may offer a low-cost alternative to current electronics-based approaches.

[1] Institute of Biomaterials and Bioengineering, Tokyo Medical and Dental University, Tokyo, Japan. [2] Kanagawa Institute of Industrial Science and Technology, Ebina, Japan. [3] Department of Molecular Medicine and Metabolism, Research Institute of Environmental Medicine, Nagoya University, Nagoya, Japan. [4] Department of Diabetes and Endocrine Medicine, Nara Medical University, Kashihara, Japan. [5] Department of Medicine and Bioregulatory Science, Graduate School of Medical Sciences, Kyushu University, Fukuoka, Japan. [6] Department of Doctor—Patient Relationships, Nara Medical University, Kashihara, Japan. [7] Research and Development Center, Medical Technology Division for Planning, Development and Marketing, Nipro Corporation, Kusatsu, Japan. [8] CAE Department, Advanced Technical Department, Nikon Systems Inc., Tokyo, Japan. [9] Metabolism and Nutrition Research Unit, Institute for Frontier Science Initiative, Kanazawa University, Kanazawa, Japan. [10] Department of Immunometabolism, Nagoya University Graduate School of Medicine, Nagoya, Japan. [11] These authors contributed equally: Akira Matsumoto, Hirohito Kuwata, Shinichiro Kimura. ✉email: matsumoto.bsr@tmd.ac.jp; suganami@riem.nagoya-u.ac.jp

Growing attention has been paid to how to prevent the development of life-threatening complications of diabetes: microvascular complications such as retinopathy, neuropathy, nephropathy, along with atherosclerotic cardiovascular diseases. For instance, clinical trials revealed that long-term intensive insulin therapy effectively suppresses the progression of diabetic microvascular complications in type 1 and 2 diabetic patients[1–3]. Nonetheless, such treatment may also increase the risk of serious hypoglycemia, which is associated with high mortality[4]. Moreover, recent studies have pointed to the role of blood glucose fluctuations such as postprandial hyperglycemia as a major etiology of diabetic complications[5,6]. Indeed, patients with similar hemoglobin A1c (HbA1c) levels, an indicator of the average blood glucose levels over the preceding few months, often show strikingly distinct patterns of blood glucose variations. A large-scale clinical study has also shown that $M$-value, an indicator of blood glucose fluctuations, is highly correlated with microalbuminuria, an early symptom of diabetic nephropathy in type 1 diabetic patients[7]. Thus, these findings came to develop a consensus that care must be taken to evaluate the daily blood glucose fluctuations, so-called "glucose spikes", as much as the commonly used fasted blood glucose levels and HbA1c, in order to maximize the efficacy of the treatment.

Despite this knowledge, an electronics-based "closed-loop" system (or artificial pancreas) represents so far the only accessible option for the patients, although it poses a number of technical issues; high cost, burdensome sensor calibration, risk of electronic failures, and so on[8,9]. Among many efforts of developing "electronics-free" or chemically driven alternatives, formulations based on glucose oxidase (GOx) and sugar-binding lectins (Concanavalin A: Con A) are the two major approaches[10–12]. However, their inherently unstable (due to the protein denaturation) and toxic nature yields in a generally short duration of function, i.e., typically a few hours. As a result, it remains to be elucidated whether these systems appropriately ameliorate the daily fluctuation of blood glucose levels in vivo.

In sharp contrast to the above, our study focuses on a "protein-free", totally synthetic approach, taking advantage of boronate-sugar binding chemistry[13–19]. Boronic acid (BA) derivatives readily complex with 1,2- and 1,3-cis-diols, including those present in glucose, through reversible boronate ester formation. Our previous studies have shown that the boronate gel-based insulin-diffusion control mechanism is weekly sustainable while also being acutely glucose-responsive on a timescale of tens of seconds[14,15,19], features potentially meeting the current unmet needs in insulin therapy, including the management of glucose spikes. We have also reported a catheter-combined device scaled suitable for mouse experiments, which, upon subcutaneous implantation, could control the glucose metabolism under both insulin-deficient and insulin-resistant conditions with at least 3-week durability[15].

In this report, we update two unique aspects to our smart gel technology that are directly relevant to the clinical translation, i.e., the scalability and the efficacy for the glucose spikes (daily blood glucose fluctuations). Being diffusion-dependent, one can expect that the power of the device, i.e., the rate of insulin release, is scalable with a linear correlation with the surface area of the gel. We prepared a hemodialysis (semipermeable) fiber-combined device by installing a thin coat of the gel throughout the fiber surface but not within, thereby achieving both a dramatically increased diffusion-active surface area (as compared to the previously reported catheter-combined type) and a smooth supply of insulin. A mathematical model was developed to characterize such device concept and to provide a quantitative basis for the scaling. We successfully obtained a tenfold scale-up in the power of the device which was suitable for treating rats, who weigh roughly ten times greater than mice. We also describe a chemical-structural modification of the gel for a temperature-independent performance, which proved to be critical for the safety in vivo. Furthermore, both the mathematical model and in vitro investigation uncovered the excellence of our device in responding to acute patterns of glucose. In accordance, we could demonstrate in vivo a marked benefit of our device in coping with the glucose spike-like symptom over a timescale of day, to our knowledge, for the first time using an electronics-free system.

## Results

### Structural optimization for minimized temperature-dependency.

A glucose-dependent shift in the equilibria of BA (between those uncharged and anionically charged: Fig. 1–i), when integrated with suitably amphiphilic acrylamide hydrogel backbone, gives rise to a reversible, glucose-dependent change in hydration (Fig. 1–ii). This hydration change is mainly caused by the change in counterions' osmotic pressure arising inside the gel in synchronization with the boronate anions[13,20]. The abrupt, rapid and the gel-surface-localized mode of (de)hydration, so-called "skin layer" formation (Fig. 1–iii), serves as an effective diffusion-switch for the gel-loaded insulin (Fig. 1–iv)[20]. We have previously established a boronate gel structure that enables the closed-loop function under physiological conditions (pH 7.4, 37 °C), accompanied by a threshold glucose-responsive concentration being around normoglycemia (100 mg/dL)[14,18]. One key aspect in the optimized chemical structure is the use of a phenylboronic acid (PBA) derivative possessing para-carbamoyl and meta-fluoro substituents (FPBA), resulting in the pKa value of 7.2, which is suitable for the glucose-sensitivity at the physiological pH (7.4).

For the remaining challenge, the gel behavior was highly dependent on the temperature (see Supplementary Fig. 1), which poses a potential safety concern; the use of a thermos-sensitive main-chain component is inevitable in order to achieve a sharp transition in hydration, i.e., skin layer formation[13,14]. For example, ethanol spray on the skin of rats markedly lowered the subcutaneous temperature (see Supplementary Fig. 2a). In this study, further modification was made to the structure with the aim of making the system practically temperature-independent in the range of clinically relevant environment (Fig. 2). To a previously reported copolymer system[14], graded amounts of N-hydoroxyethylacrylamide (NHEAAm) was introduced, which is a hydrophilic comonomer bearing hydroxyl group to serve as an intermolecular cross-linker via boronate ester formation with FPBA (Fig. 2a). The most straightforward approach to moderating the temperature-dependency of our system is to increase the fraction of boronate component (FPBA) and decrease that of thermos-sensitive NIPMAAm in feed. However, as a side-effect of doing so, thus increased boronate fraction causes an excessively increased hydrophobicity of the overall network, making it difficult to function under the physiological temperature[16,17]. We hypothesized that the presence of hydroxyl group (NHEAAm) in the structure with an optimal amount would aid in both mitigating the hydrophobicity and preventing unwanted network loosening, thanks to the ability to promote additional intermolecular cross-links (see also Supplementary Fig. 1). Careful attention was paid to the retaining of sufficient dehydration level ($Y$ axis) at lower glucose conditions so that the gel would not compromise on the ability of skin layer formation. We systemically studied on phase diagrams (Fig. 2b) the effect of NHEAAm on those factors, and came up with an optimum formulation that achieves a practically temperature-independent closed-loop function in the range of 25–45 °C (Supplementary Fig. 3).

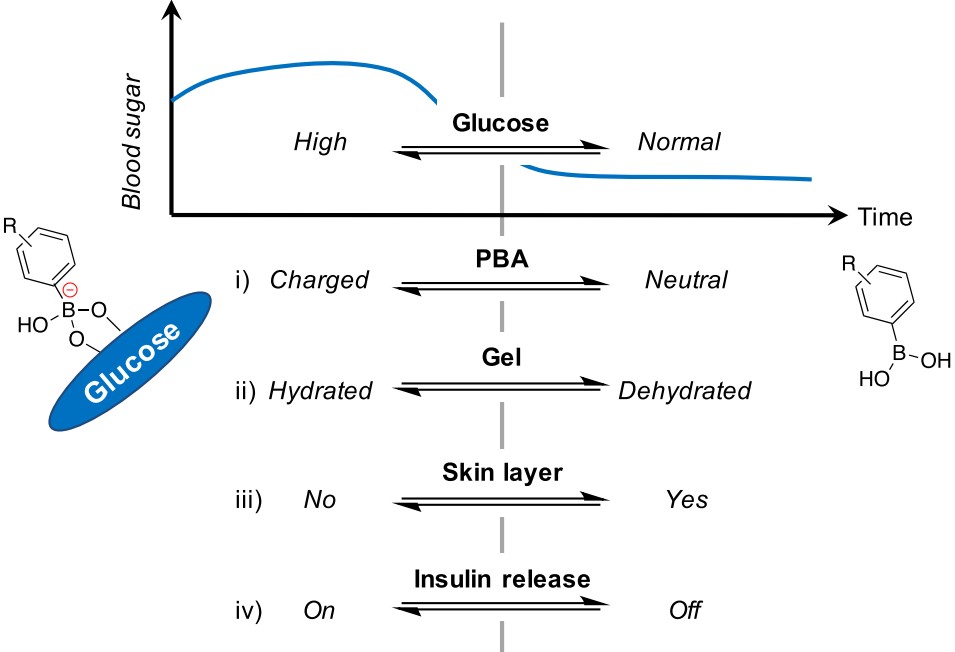

**Fig. 1 Schematic illustration of boronate gel-based closed-loop system.** (i) Glucose-dependent equilibria of boronic acids. (ii) The fraction change between electrically neutral boronic acid moiety (right) and the boronate anions (left) translates into a change in hydration of the gel due to the change in counterions' osmotic pressure. (iii) A rapidly evoking dehydration (at low glucose) results in a formation of a gel-surface-limited, microscopically dehydrated layer, so-called "skin layer". (iv) The skin layer provides a rapid, reversible and continuous diffusion control mode for the gel-loaded insulin.

**Device preparation**. The structural concept and fabrication procedure for the hemodialysis fiber-combined device is summarized in Fig. 3. A commercial hollow fiber dialyzer (FIX-110Seco: NIPRO) was first removed from the housing and disassembled into a bundle consisting of ca. 25 fibers. The perfectly (totally inspected in commercial grade) glue-sealed end (Fig. 3b: ii) was made use for the following catheter connection (Fig. 3b: iv). The gelation was carried out by first immersing the bundle in a pre-gel methanol solution dissolving all monomers and initiator at room temperature, which was then quickly transferred to a liquid paraffin bath heated at 50 °C for overnight reaction (Fig. 3b: iii). This procedure and the subsequent solvent washing process yielded a gel installation finely confined within the fiber matrix (Fig. 3c). This simple protocol proved feasible owing to the use of incompatible solvent pair, i.e., methanol and liquid paraffin, and the facts that none of the monomers is soluble in liquid paraffin phase and that a strong capillary force is inherent to the fiber matrix that helps sustain the pre-gel methanol solution within itself during the gelation, all aiding in the finely and autonomously fiber-matrix-confined gelation (Fig. 3c; see also Supplementary Fig. 4). Thus gel-installed, catheter-connected part was further glue-connected to a silicone-made insulin reservoir, sterilized by ethylene gas exposure, and filled with recombinant human insulin (Fig. 3b: v–vii). Finally, the gel-installed region was covered with a piece of cellulose dialysis membrane with the molecular weight cut-off of 12–14 kD, well above that of insulin (5807), to prevent fibrillization around the device during implantation.

**In vitro characterization**. In this study, we aimed to scale the device capability to what is suitable for rat experiments. For this purpose, we developed a mathematical model to characterize the closed-loop function of the device and to gain a quantitative basis for the scaling. The flow field surrounding the device and the insulin/glucose diffusion around/inside the device were simulated using OpenFOAM software based on finite volume method. The model was optimized in a way to best reproduce the characteristics of our release experiment results obtained by an HPLC set-up (Fig. 4d) (for details see Experimental section and Supplementary Fig. 5). A typical device geometry having 19 concentric fibers is indicated in Fig. 4a. Particularly, a 60° pie-shaped region (shown in purple color in Fig. 4a) was analyzed in detail for the time evolution of glucose and insulin distributions (Fig. 4b) for different patterns of glucose. Based on the axial symmetry, these results were integrated (into that of 360°) to predict the outflow insulin profiles that are shown in Fig. 4c in comparison to those experimentally obtained (Fig. 4d). The established model demonstrates that under the flow rate of 1 mL/min the device is able to tightly synchronize the insulin release with acute/frequent changes of glucose (Fig. 4c: left), but poorly so with less-acute patterns (Fig. 4c: middle and right) with marked decays in the release, which is consistent with the experimentally obtained results (Fig. 4d). The mathematical model result shown in Fig. 4b explains the cause of such decay, that is, with an extended exposure to hyperglycemic condition, the inflow of insulin from the reservoir into each fiber becomes insufficient over time (Fig. 4b: $t = 80$–90 min), making the response somewhat pulsatile. Figure 4b further demonstrates that the rate of insulin release, once exceeding the threshold glucose concentration (100 mg/dL), becomes almost independent of the glucose concentration, a behavior resulting from the property of the gel.

**Evaluation of glucose-responsive insulin release in vivo**. We assessed the in vivo performance of the device in healthy normoglycemic Sprague−Dawley (SD) rats. Figure 5a shows the protocol of the experimental design, in which an intraperitoneal glucose tolerance test (GTT) was performed with a glucose dose of 3.0 g/kg[2] (GTT-1) and 7 (GTT-2) days after subcutaneous implantation of the device. After 6 h of fasting, apparent hypoglycemia did not occur in healthy rats implanted with the device containing recombinant human insulin (Fig. 5b, GTT-1 and GTT-2). The GTT revealed that the human insulin-loaded device significantly ($P <$

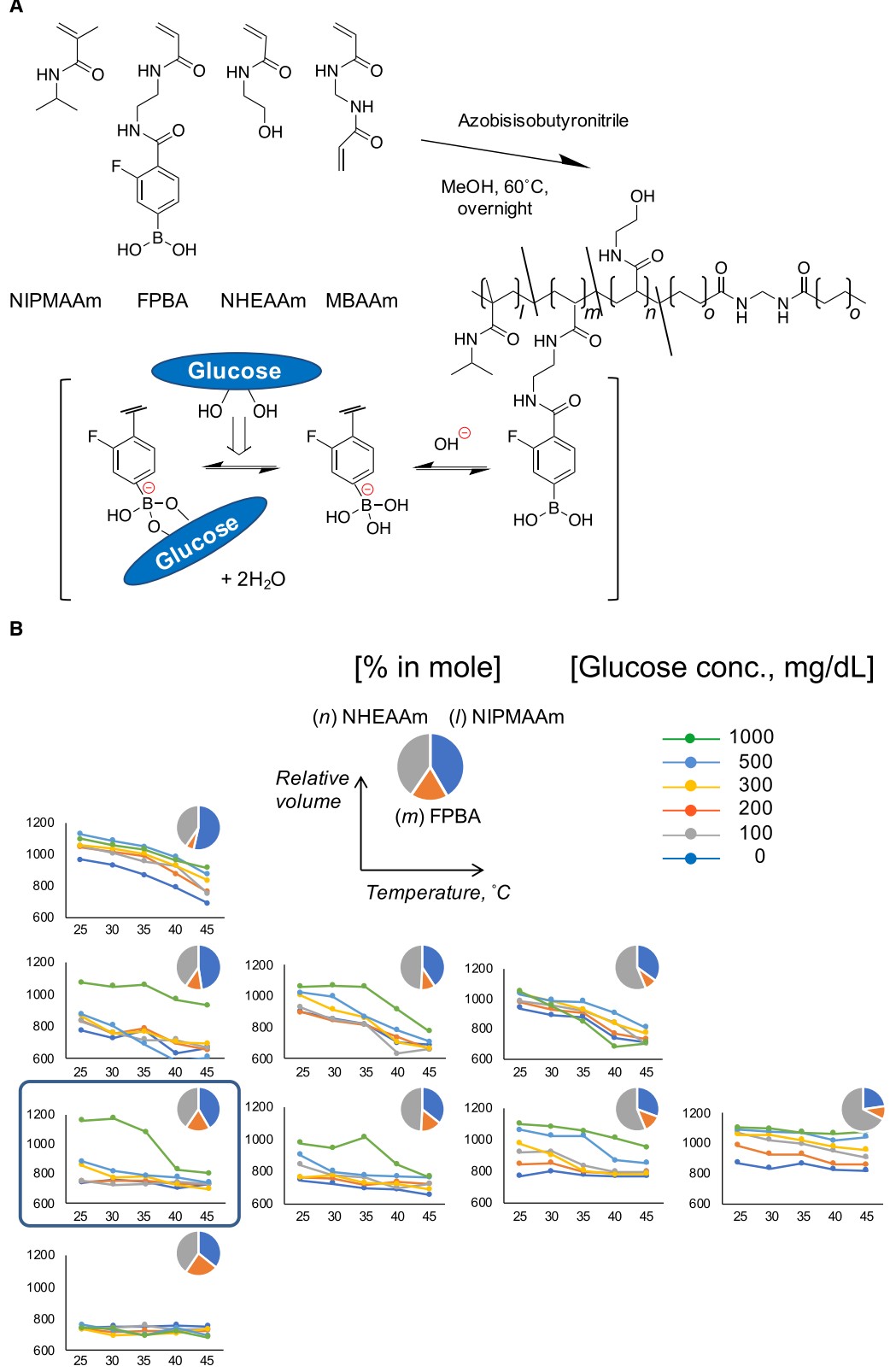

**Fig. 2 Chemical structure of smart gel with minimized temperature-dependent functionality. a** Generalized chemical structure of the gel. **b** Phase diagrams of the gel for different monomer compositions in feed showing their equilibrium volume (hydration) changes as functions of temperature and glucose concentrations investigated at pH 7.4. Pie charts indicate feed compositions of each monomer. In each formulation, the gel tends to swell (become more hydrated) with increase of glucose and decrease of temperature. It is also observed that the relative volume of the gel increases (network-loosening) with increased content of NHEAAm, and decreases with increased content of FPBA. Shown in a blue rectangle is the formulation used in this study ($l:m:n:o$ = 39.56:16.95:38.48:5.00).

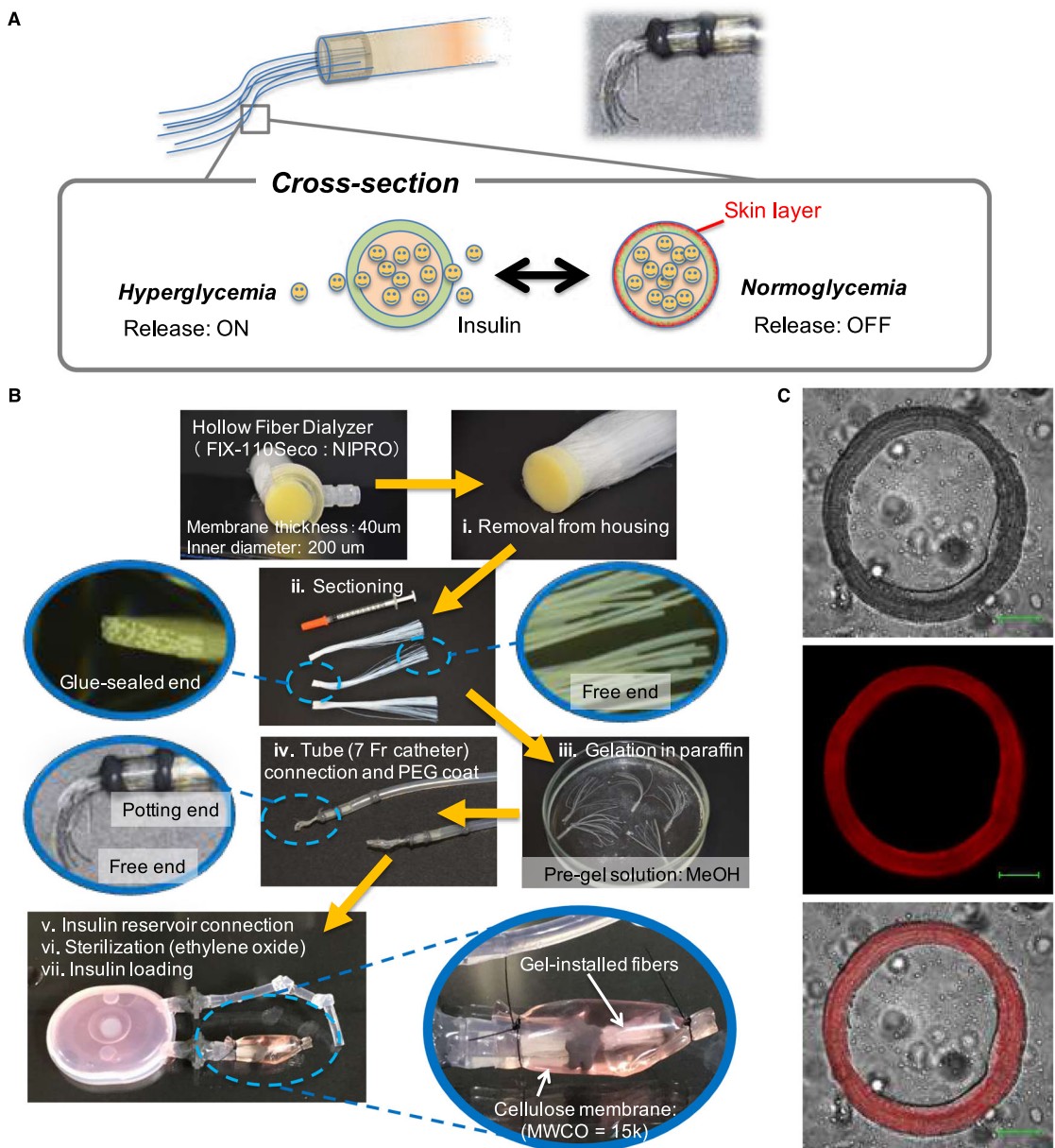

**Fig. 3 Hemodialysis fiber-combined device. a** Schematic illustration of the skin-layer-driven working principle. **b** Summary of fabrication procedure. (i) A commercial hollow fiber dialyzer was first removed from the housing and disassembled into a bundle consisting of ca. 25 fibers. (ii) The free-end of the bundle was cut in the length 15 mm. After washing and drying, gelation was carried out by first immersing the bundle in a pre-gel methanol solution dissolving all monomers and initiator at room temperature, which was then quickly transferred to a liquid paraffin bath heated at 50 °C for overnight reaction (iii). After the reaction, the bundle was cleaned and dried. (iv) The bundle was further coated with a thin layer of poly(ethyleneglycol). (v) The dried bundle was then connected to a silicone-made insulin reservoir by using polypropylene-made tube connectors, followed by waterproofing treatment by epoxy resin. A 4 French silicone catheter was connected to the other arm of the reservoir. (vi) The device was then sterilized by an ethylene oxide gas sterilizer. (vii) Finally, the reservoir was filled with recombinant human insulin, and the gel-installed region was covered with a piece of cellulose dialysis membrane which was loosely tied with silk suture. **c** Microscopic images of a 40-μm-thick gel-installed fiber section. Top: transmittance image, middle: fluorescence image of Rhodamine-conjugated gel, bottom: superposition of the top and the middle. Scale bar: 50 μm.

0.05) attenuated a transient increase in blood glucose levels compared to the phosphate-buffered saline (PBS)-loaded device (Fig. 5b, GTT-1). In response to the blood glucose levels, serum concentrations of human insulin (released from the device) were increased roughly up to those of endogenous insulin in the rats implanted with the PBS-loaded device. The endogenous insulin release (evaluated by serum C-peptide concentrations) was significantly ($P < 0.05$) reduced in the rats with human insulin treatment before and after the glucose injection. These observations suggest that basal as well as inducible insulin release from the device

are capable of reducing the burden of insulin-secreting pancreatic β-cells. Moreover, the blood-glucose-lowering effect of the device was sustained for at least 7 days (Fig. 5b, GTT-2), suggesting the functional durability of the device.

To assess the glucose-level-dependent insulin release from the device in detail, we monitored blood and subcutaneous glucose levels along with serum human insulin concentrations in rats receiving continuous glucose infusion (Fig. 6a). We confirmed that the subcutaneous glucose levels were almost comparable to the blood glucose levels without appreciable delay during the

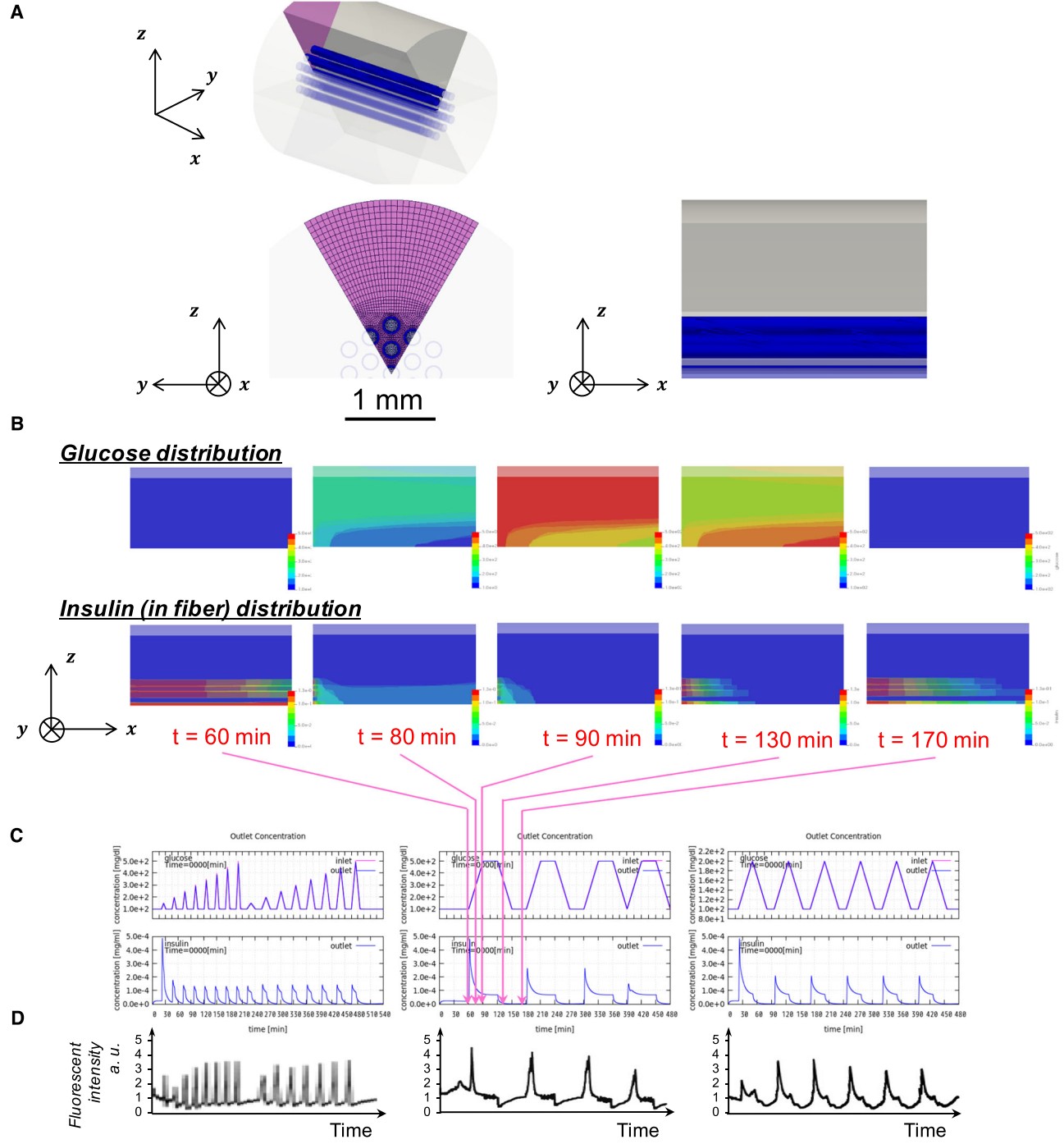

**Fig. 4 Mathematical modeling. a** Analyzed field, an approximated geometry of the device. **b** Snapshots of glucose (top) and insulin (bottom) distributions when challenged with glucose pattern shown in (**c**) (middle). **c** Simulated outflow insulin release profiles (bottom) when challenged with different glucose patterns (top). **d** Experimentally obtained insulin release profiles using an HPLC set-up.

experimental period (Fig. 6b). In this setting, serum concentrations of human insulin rose approximately 20 min after the blood glucose levels increased, and declined 30 min after those decreased (Fig. 6c). This process is composed of the following steps: the implanted device senses the subcutaneous glucose levels surrounding the device, the device starts releasing human insulin into the subcutaneous blood flow, and the human insulin reaches the systemic circulation. These findings suggest that the subcutaneously implanted device promptly responds to the changes in blood glucose levels. Our device has a practically temperature-independent closed-loop function in the range of

25–45 °C (Supplementary Fig. 3). We confirmed that this temperature-independency under physiological conditions was critical in vivo to prevent an undesired burst release of insulin immediately after implantation of the device (Supplementary Fig. 2b). We further evaluated the biosafety of the device 9 days after implantation in healthy normoglycemic rats (Fig. 5a). Histological examinations using hematoxylin and eosin staining revealed only mild inflammatory cell infiltration around the device (Supplementary Fig. 6a). Implantation of the device did not show significant ($P > 0.05$) changes in the population of the number of white blood cells in the blood and serum CRP

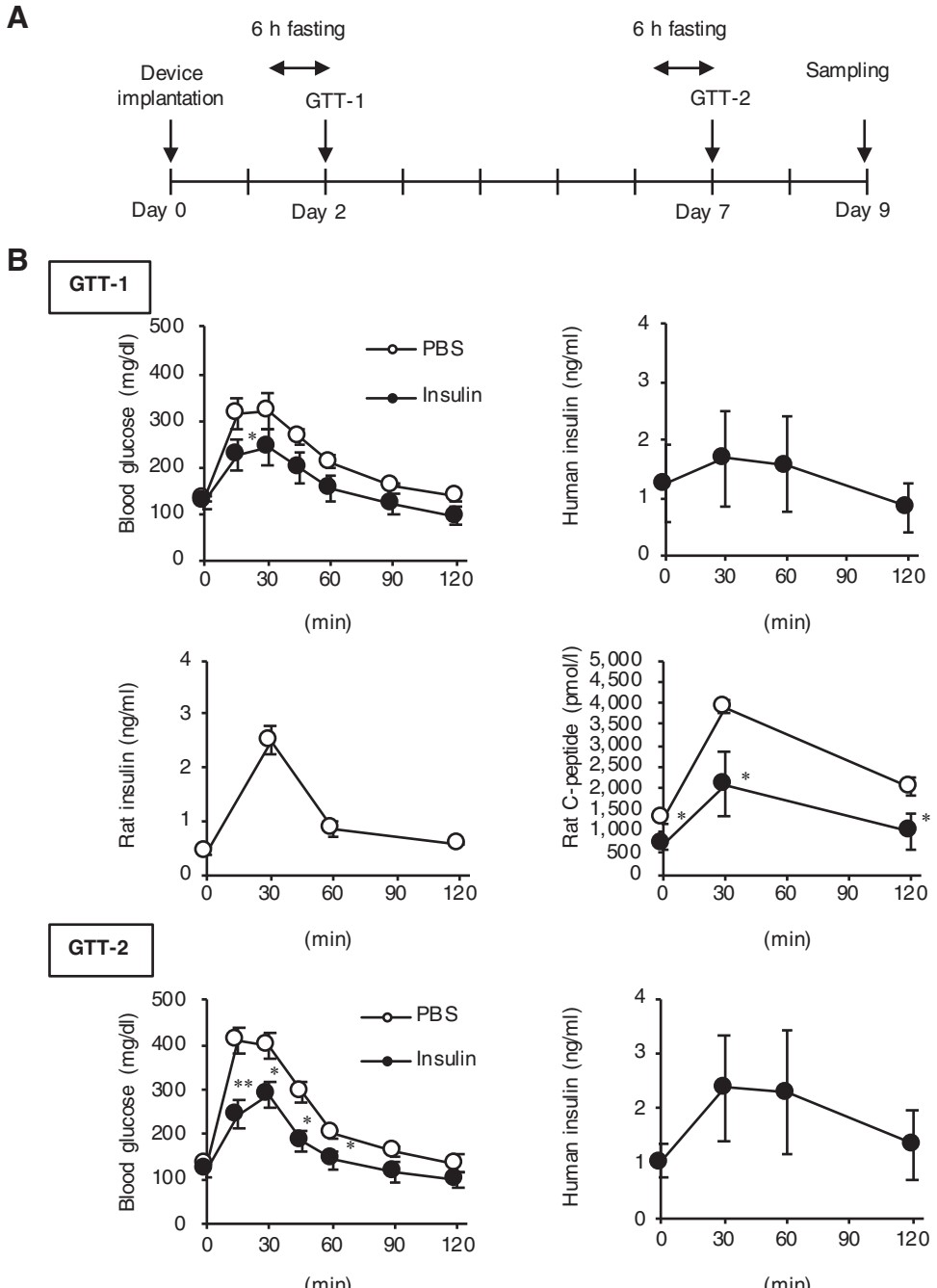

**Fig. 5 Blood-glucose-lowering effect of the device in healthy normoglycemic rats. a** Protocol of the experimental design using healthy normoglycemic rats. **b** Glucose tolerance test (GTT). GTT was conducted on days 2 (GTT-1) and 7 (GTT-2) after the device containing PBS or human insulin was implanted subcutaneously in the back skin of healthy normoglycemic rats. Glucose (3 g/kg body weight) was injected intraperitoneally to the rats after the 6-h fast. Concentrations of blood glucose, serum human insulin (derived from the device), serum rat insulin (endogenous), and serum rat C-peptide were measured at the indicated time points. *$P < 0.05$ and **$P < 0.01$ vs. the PBS group. $n = 5$–7.

concentrations (Supplementary Fig. 6b). Of note, in our previous study on the catheter-combined device, colony formation assay was carried out using Chinese hamster lung fibroblasts (V79). The medium prepared from the culture of the device did not affect colony formation of V79 cells, suggesting no detectable cytotoxicity of the gel itself[15].

**Therapeutic efficacy for type 1 diabetic rats**. We then assessed the therapeutic effect of the device on glucose metabolism in streptozotocin (STZ)-induced type 1 diabetic rats (conditions

with absolute insulin deficiency). Figure 7a shows the protocol of the experimental design, in which the rats were implanted with the device containing PBS or human insulin 7 days after STZ injection, and observed for additional 7 days. The human insulin-loaded device effectively reduced ad lib blood glucose levels along with the continuous insulin release up to day 7 (Fig. 7b). In line with these observations, the human insulin-loaded device markedly suppressed the increase in HbA1c levels and the amount of water intake in type 1 diabetic rats (Fig. 7c, d). These observations suggest that the device is stably controlling glucose metabolism under insulin-deficient conditions for at least 7 days.

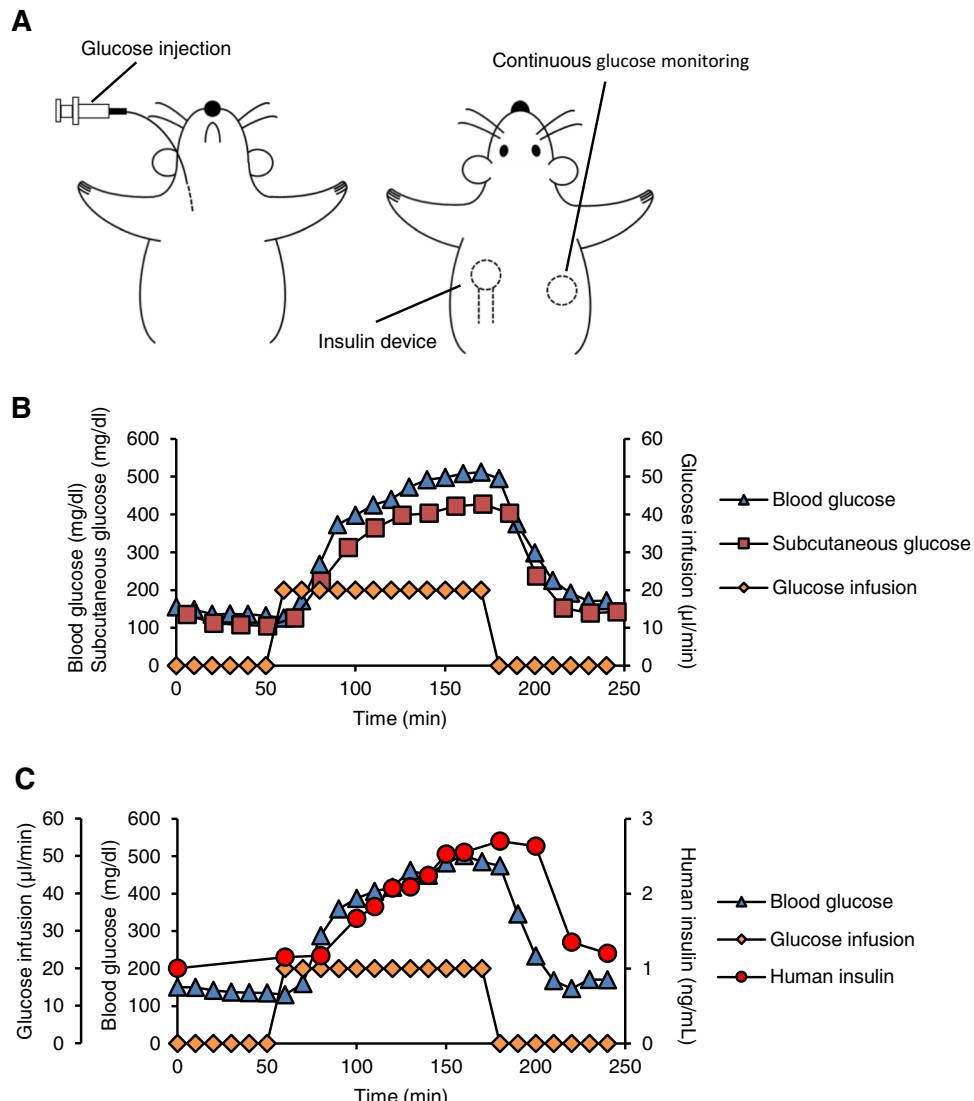

**Fig. 6 Glucose-level-dependent insulin release from the device in vivo. a** Schematic illustration of the experimental design. The device containing human insulin was subcutaneously implanted along with a continuous glucose monitoring system in the back skin of healthy normoglycemic rats. The rats received continuous glucose injection through the right internal jugular vein for the indicated period. **b** Time course of blood and subcutaneous glucose levels. Glucose solution was continuously injected from 60 to 180 min. **c** Time course of blood glucose levels and serum human insulin concentrations. Glucose solution was continuously injected from 60 to 180 min.

**Amelioration of glucose fluctuation in mild diabetic rats**. Finally, we evaluated the effect of the device on daily blood glucose fluctuations, since accumulating evidence has suggested that blood glucose fluctuations such as postprandial hyperglycemia would be a major cause of diabetic complications such as atherosclerosis. The onset of diabetes begins with postprandial hyperglycemia due to abnormal pancreatic islet cell function, without elevation of fasting blood glucose levels[21]. Therefore, we intraperitoneally injected low-dose STZ (30 mg/kg body weight) to SD rats to induce mild diabetes (Fig. 8a). The subcutaneous glucose levels were monitored using a continuous glucose monitoring system before and after implantation of the human insulin-loaded device. Before the implantation, the rats clearly exhibited nocturnal increase in the subcutaneous glucose levels (up to 200 mg/dL), whereas the levels were approximately 150 mg/dL in the diurnal period (Fig. 8b, c), offering a suitable model to verify the effect of the device on blood glucose fluctuations and the safety against hypoglycemia. In this experiment, the fluctuations of blood glucose levels were markedly reduced after the

implantation; i.e., there was a large decrease in the subcutaneous glucose levels in the nocturnal period, while the levels were almost stable in the diurnal period. These findings were supported by the M-value, an indicator of daily blood glucose fluctuations (Fig. 8d).

## Discussion
Being synthetic, our smart gel technology is highly stable, potentially free from immunotoxicity issue. Owing to the skin layer-based diffusion-dependent mechanism, it represents so far the only known chemical system accomplishing both at once a weekly durability and a remarkably acute response in a matter of tens of seconds. Toward clinical translation, we asked a fundamental question regarding the scale-up in the power of the device. In principle, under the assumption of a constant insulin concentration gradient, the rate of insulin release is scalable with a linear correlation with the surface area of the gel. To prove this concept in a reasonably compact device dimension, we tested the

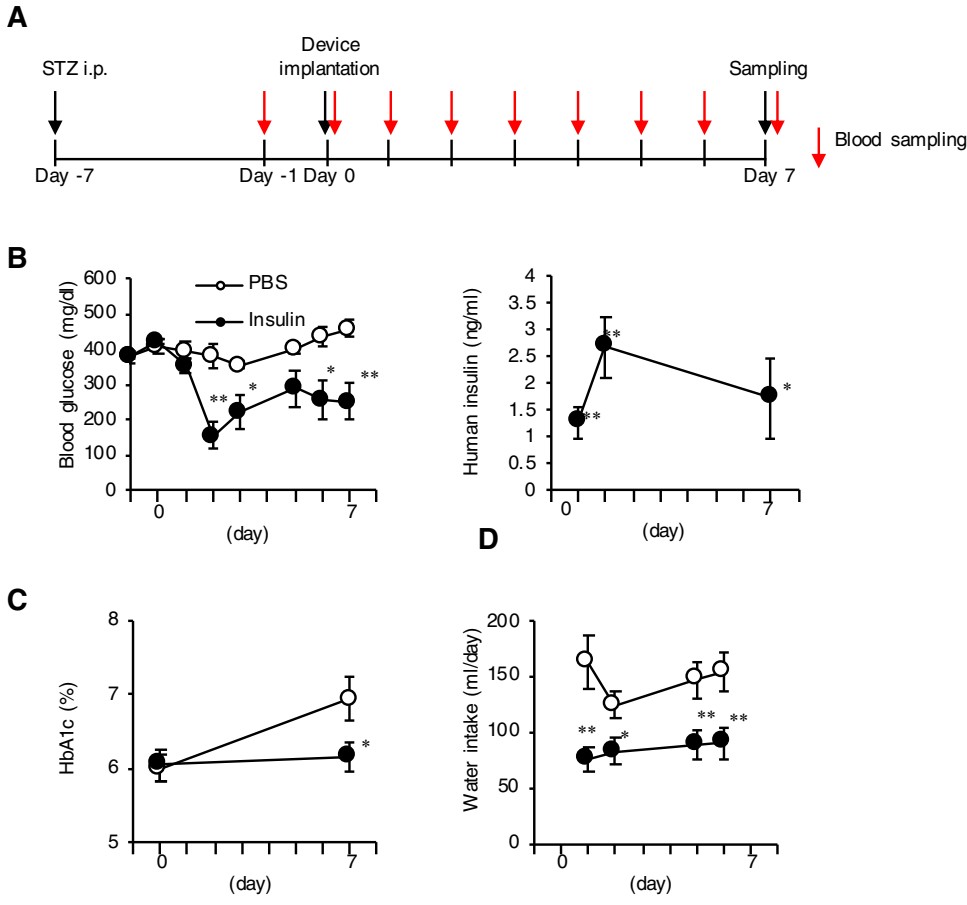

**Fig. 7 Therapeutic efficacy for type 1 diabetes in rats. a** Protocol of the experimental design. Type 1 diabetes was induced by the intraperitoneal injection of streptozotocin (STZ, 40 mg/kg body weight) on day −7, and then received subcutaneous implantation of the device containing PBS or human insulin on day 0. **b** Time course of blood glucose levels and serum human insulin concentrations under ad lib fed conditions. **c** HbA1c levels before and 7 days after implantation of the device. **d** Time course of the amount of daily water intake. *$P < 0.05$ and **$P < 0.01$ vs. the PBS group. $n = 6$–8.

idea of hemodialysis fiber-combined device. The calculated diffusion-active surface area for this device is about 188 mm$^2$, which is over a hundred times greater than that of the previously reported catheter-combined device (1.5 mm$^2$). The present study demonstrates that ten times scale-up (as compared to the catheter-combined device) is achievable based on such strategy, which proved to be suitable for rat experiments. In the meantime, our mathematical model revealed that the inflow of insulin coming from the reservoir into each fiber was a rate-determining factor, indicating that there is still room for the scale-up, potentially by another ten times. The resulting lag time of the insulin inflow made the device behave "pulsatile", thus posing a challenge to synchronize with somewhat continuously hyperglycemic patterns (Fig. 4). This behavior may be regarded as a "side-effect" of the scale-up of the device, on the one hand, but such pulsatile-response aided in coping with the glucose-spike-like symptom, on the other. The insufficiency of the insulin supply may be solved by additional engineering involving the reservoir-internal pressure and flow controls. In terms of shortening the diffusion distance for insulin, a microneedle-based formulation, bearing an array of short needles ranging <1 mm in length, may solve the situation[19,22]. One can also consider the use of a concentrated type of insulin, which has been in clinical use, for additional scale-up.

Since a variety of medications for diabetes are currently available in clinical practice, it is important to prioritize their preventive effect on diabetic complications. In this regard, substantial evidence has shown that hypoglycemia and blood glucose fluctuations are

key determinant factors, which increase the risk of diabetic complications possibly through oxidative stress and proinflammatory cytokines[23]. To the best of our knowledge, our device, for the first time as an electronics-free system, has remarkably ameliorated daily blood glucose fluctuations under diabetic conditions in vivo over timescale of day, without inducing apparent hypoglycemia. This is reminiscent of the superiority of continuous subcutaneous insulin infusion therapy (CSII) over conventional insulin therapy with multiple injections; i.e., CSII markedly improves blood glucose fluctuations and reduces the development of complications in type 1 diabetic patients compared to conventional insulin therapy[24–26]. Most recently, CSII combined with real-time continuous glucose monitoring, known as sensor-augmented pump therapy (SAP), has been introduced to the market, with even higher efficacy than standard CSII therapy[27,28]. One limitation of these systems is that the postprandial insulin injection has to be manually regulated. In contrast, insulin release from our device includes the glucose-independent baseline release and the glucose responsive inducible one, both of which are chemically controllable on the basis of the "smart gel" technology[15]. Further noteworthy is that the threshold glucose-responsive concentration, above which the insulin release is induced, can also be chemically tailored[14,17]. Since another advantage of our device is long-term durability (~1 week), it is interesting to investigate the preventive effect of our device on the development of diabetic complications in animal models.

Similar to SAP, our device is affected by a temporal gap (~10 min) between interstitial and intravascular glucose levels in the postprandial state[29,30]. Such lag time must be taken into

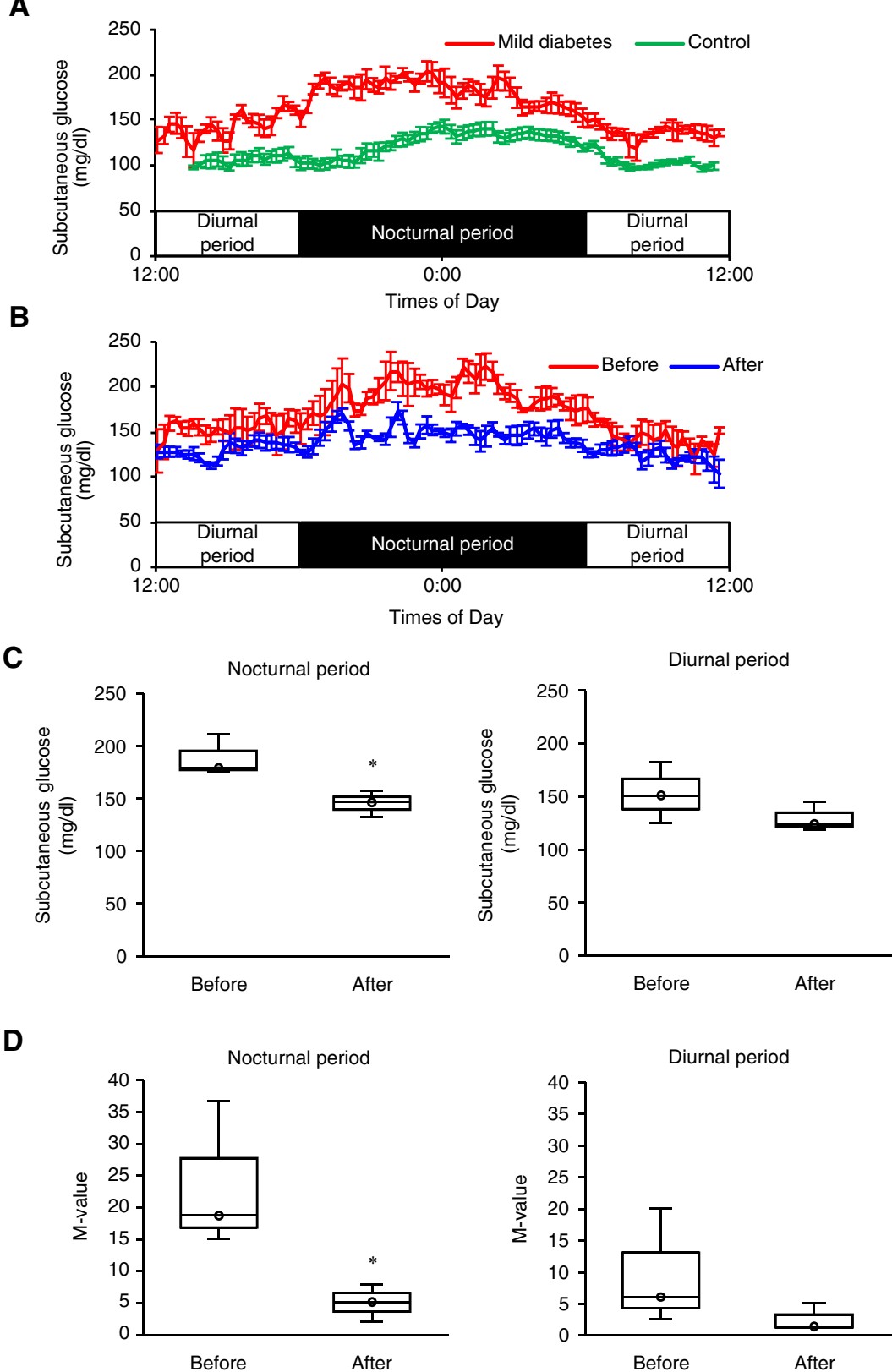

**Fig. 8 Amelioration of glucose fluctuations in mild diabetic rats.** Mild diabetes was induced by intraperitoneal injection of low-dose STZ (30 mg/kg body weight). Then, the rats received subcutaneous implantation of the device containing human insulin along with a continuous glucose monitoring system. **a** Daily glucose fluctuations in control and mild diabetic rats. Mild diabetic rats showed elevation of blood glucose levels mainly in the nocturnal period (active phase). **b** Daily glucose fluctuations before and after implantation of the device. Treatment of the device effectively suppressed the nocturnal increase in blood glucose levels. **c** Mean subcutaneous glucose levels in the diurnal and nocturnal periods. Treatment of the device (**d**). Evaluation of daily glucose fluctuations indicated by the *M*-value. *$P < 0.05$ vs. before implantation. $n = 3$.

consideration, in which the device-derived insulin reaches the systemic circulation (~15 min)[31,32]. In this study, we provide evidence on in vivo pharmacokinetics of the device-derived insulin. From the viewpoint of clinical translation, it is critically important to validate the glucose-responsive insulin release from our device in vivo, in addition to in vitro evaluation. Our results strongly supported the "skin layer"-regulated rapid on−off switching behavior in vivo. As revealed by the mathematical modeling study, subcutaneous microcirculation potently influences insulin pharmacokinetics. Although our new gel formulation makes the performance stable in the temperature range of 25–45 °C (Supplementary Fig. 3), the temperature change would affect subcutaneous microcirculation. For the next step, we need to assess the impact of temperature change on insulin release when we translate our device into clinical practice.

Considering the clinical application of our device, it is necessary to verify the ability to release insulin in vivo. On the basis of the formula obtained in in vitro experiments (Fig. 4), we can estimate a typical daily dose released from our device as 6 mU/kg body weight/day under the static condition and 1.2 U/kg body weight/day under the flow rate of 1 mL/min (Supplementary Fig. 7), assuming the patient's body weight to be 50 kg. On the other hand, the average dose of insulin generally required for treating human diabetic patients is about 0.7 U/kg body weight/day[33]. Since the flow rate of 1 mL/min in the subcutaneous tissue around the device is not plausible in vivo, we are not optimistic that our current device can be applied directly clinically. Accordingly, for future directions, we should evaluate the microcirculation around the implanted device to estimate the insulin release rate in vivo more precisely. Nevertheless, we consider that clinical applications will be possible because the efficacy of our device can be modulated by factors such as insulin concentration and the number of fibers.

In conclusion, we have proven the safety and efficacy of our boronate gel-based closed-loop device for rats, who weigh roughly ten times greater than mice, a model previously investigated. In accordance with mathematical modeling and in vitro studies, the device showed a marked efficacy to mitigate the glucose-spike-like symptom for at least 24 h, to our knowledge, for the first time using an electronics-free system in vivo. With diffusion-dependent scalability along with unique combination of weekly durability and acute responsiveness (in a matter of tens of seconds), this technology may offer a remarkably economic alternative to the current electronics-based closed-loop systems to treat diabetes and glucose spikes.

## Methods

**Reagents.** N-isopropylmethacrylamide (NIPMAAm), 4-(2-acrylamidoethylcarbamoyl)-3-fluorophenylboronic acid (AmECFPBA), N-hydroxyethylacrylamide (NHEAAm), N,N′-methylenebisacrylamide (MBAAm), 2,2′-Azobis(2,4-dimethyl-valeronitrile) (V-65), 2,2′-azobisisobutyronitrile (AIBN), FITC (fluorescein isothiocyanate)-labeled bovine insulin and all organic solvents (ethanol, methanol, hexane, dimethyl sulfoxide (DMSO)) were all purchased from Wako Pure Chemical Industries (Tokyo, Japan). Acryloxyethyl thiocarbamoyl rhodamine B was purchased from Polyscience, Inc. (Warrington, USA). NIPMAAm was recrystallized in hexane and then dried in vacuo overnight before use.

**Preparation of gels.** Cylindrically shaped "capillary" gels were used for the study shown in Fig. 2b and Supplementary Fig. 1. The gels were prepared in a 1 mm diameter glass capillary (Calibrated Pipets 200 μL: Drummond Scientific Company, Broomall, PA, USA) by radical copolymerization under argon atmosphere, using AIBN as an initiator in the presence of MBAAm as a crosslinking agent in DMSO. The concentrations of total monomers, AIBN and MBAAm in feed, were 2.5 M, 7.5 mM and 125 mM, respectively. The obtained capillary gels were thoroughly washed with milli-Q water to remove any unreacted compounds, and finally conditioned in each buffer solution for at least overnight.

**Preparation of hemodialysis fiber-combined device.** A commercial hollow fiber dialyzer (FIX-110Seco: NIPRO Corp., Osaka, Japan) was first removed from the

housing and disassembled into a bundle consisting of ca. 25 fibers. The free-end of the bundle was initially cut in the length 15 mm. The bundle was washed with EtOH in sonication for 15 min and then dried. The gelation was carried out by first immersing the bundle in a pre-gel methanol solution dissolving all monomers and initiator (V-65) at room temperature, which was then quickly transferred to a liquid paraffin bath heated at 50 °C for overnight reaction. After the reaction, the bundle was cleaned by paper towel to remove the remaining liquid paraffin and then washed by sonication, sequentially in MeOH (for 10 min at room temperature, twice), milli-Q water (for 30 min at room temperature, twice), and finally in PBS (for 30 min at 4 °C, twice). The free-end of the bundle was further coated with a thin layer of poly(ethyleneglycol) (Tetragel) in the same procedure previously reported[15]. Two types of four-arm branched (tetra) poly(ethyleneglycol) macromers of 10,000 in Mw (2500 for each chain) bearing either amino (PTE-100PA, NOF Corp.) or active ester (N-hydroxysuccinimide-activated) groups at each four end (PTE-100CS, NOF Corp.) were dissolved separately in PBS (1 g/20 mL) adjusted to be pH 7.4 and pH 5.8, respectively. These were mixed in a stoichiometric ratio and quickly homogenized at room temperature, to which the free-end part was gently immersed and then kept overnight under high-humidity atmosphere. The dried bundle was then connected to a silicone-made insulin reservoir (see Supplementary Fig. 8 for detailed dimension) by using polypropylene-made tube connectors (Mini-fitting Connector VFI 126; ISIS Co., Ltd., Osaka, Japan), followed by waterproofing treatment by epoxy resin (Shin-Etsu Silicone 1 Component RTV (KE-3424-G): Shin-Etsu Chemical Co. Ltd.). A 4 French (1.2 mm of diameter) silicone catheter (Access Technologies, Skokie, IL, USA) was connected to the other arm of the reservoir (see Supplementary Fig. 7) in the same way as described above with the exception of the type of connector (Mini-fitting Connector VFI 116; ISIS Co., Ltd., Osaka, Japan). Thus, prepared device was sterilized by an ethylene oxide gas sterilizer (Cartridge type fully automatic ethylene gas sterilizer SA-H360, Canon Lifecare Solutions. Inc., Tokyo, Japan) for 4 h at 40 °C, followed by over 15-h aeration. The device was then filled with recombinant human insulin (Humulin R: Eli Lilly Japan K. K., Kobe, Japan). Finally, the gel-installed region was covered with a piece of cellulose dialysis membrane with the molecular weight cut-off of 12–14 kD (Spectra/Por Dialysis Membrane Standard RC Tubing, MWCO: 12–14 kD; Fisher Scientific, MA, USA) and loosely tied with silk suture.

**Confocal imaging.** The fiber-installed gel section was examined by confocal laser scanning microscopy as shown in Fig. 3c. For this experiment the gelation was carried out in the presence of small amount of acryloxyethyl thiocarbamoyl rhodamine B for the gel-matrix-specific staining. For comparison, a fiber immersed in a solution containing the same amount of rhodamine as above but no monomer was prepared. Fibers immersed in optimum cutting temperature (OCT) compound (OCT Compound 4583, VWR International LLC) were placed vertically in cryomold (Tissue-Tek Cryomold Biopsy (15 × 15 × 5 mm), VWR International LLC). Then, the suspensions in cryomold were placed on a freezing microtome sample holder (Leica CM1950, Leica Biosystems Inc.) for freezing. Obtained frozen suspensions were sectioned with a thickness of 40 μm using a freezing microtome (Leica CM1950, Leica Biosystems Inc.) and then washed ten times with Milli-Q water and methanol to remove the OCT compound and excess rhodamine. After washing, rhodamine-stained gel in the sections were analyzed on a confocal laser scanning microscopy (C2 + confocal microscope, Nikon Co. Ltd.) using the imaging software (NIS-Elements, Nikon Co. Ltd.). For rhodamine, the 561-nm line of the confocal laser was used with optimized value (laser power: 20, zero photomultiplier tube module (PMT) offset: 0, PMT HV: 120). The control sample (immersed in rhodamine in the absence of monomers) showed no staining after the washing, thus confirming the gel-matrix-specific rhodamine-staining as shown in Fig. 3c.

**Release experiments.** Release experiments shown in Fig. 4d and Supplementary Fig. 3 were carried out using an HPLC system (JASCO, Tokyo, Japan) equipped with two pumps and internal detectors for refractive index and fluorescence intensities (Supplementary Fig. 5), in an identical manner previously reported[14]. The device loaded with 130 mg/L of FITC-labeled bovine insulin was put in a Tricorn Empty High-Performance Column (GE Healthcare, Chicago, IL, USA) having dimensions of 10 and 50 mm in inner diameter and length, respectively. The column was settled in a thermo-stated (at 37 °C) flow of PBS (pH 7.4, 155 mM NaCl) containing glucose (100 mg/dL), the flow rate of which was constantly 1 mL/min in a chamber connected to the HPLC system. The initial equilibration was allowed until no leakage of insulin from the device was observable, which typically took a few hours. The fluorescence intensity of the solution at 520 nm (excitation wavelength: 495 nm) was monitored to trace the released amount of FITC-labeled insulin from the device. Two solutions, PBS with and without glucose (1000 mg/dL), were prepared and supplied through the two pumps of the system. The mixing-fraction of the two pumps was continuously controlled by an equipped software (ChromNAV, JASCO) to provide desired temporal gradient patterns of the glucose concentration between 100 and 500 mg/dL. In situ glucose concentration was monitored at the chamber outflow by the RI detector throughout the experiment.

Release experiment under static (no flow) condition shown in Supplementary Fig. 7 was performed as follows. Each device loaded with recombinant human insulin (Humulin R: Eli Lilly Japan K. K., Kobe, Japan) or rapid-acting insulin (APIDRA R: Sanofi Japan K. K., Tokyo, Japan) was independently immersed in a sterilized tube container filled with 10 mL PBS with either 100 or 300 mg/mL

glucose, and kept at 37 °C in an incubator throughout the experiment. At time points depicted in Supplementary Fig. 7, PBS was exchanged with freshly prepared ones. The collected PBS samples were stored at 4 °C until measurement. Insulin concentration in the collected PBS samples were measured using a commercial enzyme-linked immunosorbent assay kits (Mercodia, AB, Uppsala, Sweden). One milliliter of each sample (or those diluted with PBS when needed) was analyzed on a 96-well plate using a Plate reader (Tecan infinite 200, Tecan Japan Co., Ltd., Kawasaki, Japan). Based on these measurements, i.e., release amounts during each time interval, the cumulative release kinetics were drawn.

**Mathematical modeling**. The flow field in the column and the insulin/glucose diffusion rate was analyzed and stimulated by OpenFOAM software, which is commonly utilized for general-purpose fluid analysis based on finite volume method. The flow field in the column was discretized with the finite volume method and analyzed by the SIMPLE method using momentum conservation equation of incompressible fluid in steady state (Eq. 1) and continuity equation (Eq. 2) as base equations.

$$(U \cdot \nabla)U = -\frac{1}{\rho}\nabla p + \nu\nabla^2 U - \frac{\nu}{k_{(x)}}U, \tag{1}$$

$$\nabla \cdot (U) = 0, \tag{2}$$

$U$ is flow speed, $p$ is pressure, $\rho$ is density, $\nu$ is kinematic viscosity coefficient, and $k(x)$ is permeability, which is defined only inside the hollow fiber tube wall.

The time-dependent insulin and glucose diffusion was analyzed by Eq. 3, and discretized by the implicit Euler method and the finite volume method. In this equation, diffusion of insulin and glucose were considered independent of the solvent without mutual interaction.

$$\frac{\partial\phi}{\partial t} + f_{(x,\phi)}\nabla \cdot (\phi U) = \nabla \cdot \left(D_{(x,\phi)}\nabla\phi\right), \tag{3}$$

$t$ is time, $\varphi$ is concentration of glucose or insulin, $D(x, \varphi)$ is the diffusion coefficient, $f(x, \varphi)$ is the permeability coefficient.

The diffusion coefficient $D(x, \varphi)$ was used for the value outside the hollow fiber tube wall, whereas inside the hollow fiber, the value was differentiated depending on the presence of the skin layer. The permeability coefficient $f(x, \varphi)$ represents the ease of advection; $f(x, \varphi) = 1$ indicates that the solute advects completely with the solvent, whereas $f(x, \varphi) = 0$ indicates that no advection occurs regardless of the flow rate. $f(x, \varphi)$ was defined only inside the hollow fiber tube wall, and different values were used depending on the presence of the skin layer. In every analysis grid, the presence of skin layer was judged from the glucose concentration in the immediate preceding time point. The diffusion coefficient and permeability coefficient were set in each analysis grid according to the existence of the skin layer. The time evolution of insulin and glucose advection diffusion equation were calculated based on the concentration distribution of insulin and glucose at the current times.

**Animals**. Six-week-old male SD rats were purchased from Japan SLC (Shizuoka, Japan). They were maintained in a temperature-, humidity-, and light-controlled room (12-h light/dark cycles) and allowed free access to water and standard chow (339 kcal/100 g, 4.5% energy as fat; CE-2, CLEA Japan, Tokyo, Japan). Type 1 diabetes was induced by intraperitoneal injection of STZ (30 or 60 mg/kg body weight) as described[15]. The device containing 1.4 mL of recombinant human insulin or PBS was implanted subcutaneously in the back skin of the rats. All in vivo experiments in this study were approved by the Animal Care and Use Committee of Nagoya University (approval number 18235).

**Glucose tolerance test and blood analysis**. The GTT was performed as described after 6-h fast[15]. Blood glucose concentrations were measured at 0, 15, 30, 45, 60, 90 and 120 min after glucose injection (3.0 g/kg body weight) by a blood glucose test meter (Glutest Mint Sanwa-Kagaku, Nagoya, Japan). Serum concentrations of human as well as rat insulin and C-peptide were measured at 0, 30, 60, and 120 min using commercially available enzyme-linked immunosorbent assay kits (Mercodia, AB, Uppsala, Sweden). HbA1c was measured using Quo-Lab (Nipro, Osaka, Japan). The population of circulating white blood cells was measured by microscopic examination of peripheral blood (Hokenkagaku-Nishinihon, Kyoto, Japan).

**Glucose level-dependent insulin release from the device in vivo**. Under isoflurane anesthesia, an intravenous catheter was implanted into the right carotid vein of SD rats. After a recovery period, 50% glucose solution was continuously infused through the catheter (20 μL/min) for 2 h. Concentrations of blood glucose and serum human insulin (derived from the device) were measured at the indicated time points in Fig. 6. Subcutaneous glucose concentrations were monitored using a flash continuous glucose monitor (FreeStyle Libre Pro; Abbot, Chicago, IL). The body temperature was also monitored with a microchip transponder system (IMI-1000; Bio Medic Data Systems, Maywood, NJ). We confirmed that the body temperature was kept stable using a heat pad during the experimental period.

**Glucose fluctuation in mild type 1 diabetic rats**. Mild type 1 diabetes was induced by intraperitoneal injection of STZ (30 mg/kg body weight). After

2 weeks, a flash continuous glucose monitor was implanted subcutaneously in the back skin of the rats, and subcutaneous glucose concentrations were monitored. Then, the rats received subcutaneous implantation of the device containing human insulin. Daily glucose fluctuations were evaluated by the $M$-value, calculated at the basis of 100 mg/dL, before and after implantation of the device, as described previously[34].

**Histological analysis**. The skin and subcutaneous tissue including the implanted device was fixed with neutral-buffered formalin and embedded in paraffin as described[15]. Sections (5-μm-thick) were stained with hematoxylin and eosin.

**Statistics and reproducibility**. Data are means ± SEM, and $P < 0.05$ was considered statistically significant. Statistical analysis was performed using analysis of variance (ANOVA), followed by the Tukey−Kramer test. Unpaired $t$ test was used to compare two groups.

**Reporting summary**. Further information on research design is available in the Nature Research Reporting Summary linked to this article.

## Data availability
The source data presented in figures and supplementary figures are available in Supplementary Data 1. Remaining information is available from the corresponding author upon reasonable request.

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

## Acknowledgements
We thank Y. Shimada and T. Sawamura (Nikon Systems Inc.) for technical support in mathematical modeling and analysis, Y. Kageyama (Kageyama Kogyo Inc.) for technical support in preparation of insulin reservoir and Center for Animal Research and Education (CARE), Nagoya University for support on animal experiments. This work was supported in part by grants-in-aid for scientific research from the Ministry of Education, Culture, Sports, Science and Technology of Japan (MEXT), the Cooperative Research Project of Research Center for Biomedical Engineering (MEXT), Program for Building Regional Innovation Ecosystem (MEXT), Japan Science and Technology Agency (JST) COI Grant Number JPMJCE1305, and Japan Agency for Medical Research and Development (Acceleration Transformative Research for Medical Innovation program). This work was also supported by research grants from the Secom Science and Technology Foundation, the Mochida Memorial Foundation for Medical and Pharmaceutical Research, the Japan Diabetes Foundation, Japan Diabetes Society Junior Scientist Development Grant supported by Novo Nordisk Pharma, Japan Foundation for Applied Enzymology, Ube Industries Foundation, Ichihara International Scholarship Foundation.

## Author contributions
A.M., H.K., S.K., H.M., K.O., Y.M.-o., T.N., A.W., M.T., and T.M. performed most of the experiments and analyzed the data. H.Y., H.Ishii, T.M., S.C., T.B., H.Y., H.Inoue, Y.O., Y.M. contributed to the design and interpretation of the study. A.M. and T.S. conceived and coordinated the research, contributed to the discussion, and wrote the manuscript. All authors reviewed the manuscript.

## Competing interests
A.M. and Y.M. are inventors on two patents related to this work filed by the National Institute for Materials Science (PCT/JP2010/073544, filed on 27 December 2010 and published on 13 July 2016; and PCT/JP2011/061869, filed on 24 May 2011 and published on 12 July 2017). A.M., H.M., T.S., M.T., and Y.M. are inventors on a patent application related to this work filed by the Tokyo Medical and Dental University (PCT/JP2016/081407, filed on 24 October 2016). All other authors declare that they have no competing interests.
