## [Peer Review File · Communications Biology]

Reviewers' comments:

Reviewer #1 (Remarks to the Author):

This manuscript by Akira Matsumoto and colleagues is extremely well-done, thorough, and highlights a previously unappreciated role for electronics-free insulin delivery system. It is novel and with some modest revisions is likely to be of interest to a broad readership.

Fig.1 should be either moved to Supplemental or results, currently, it is in the introduction. The introduction section is too lengthy. My advice is to make it short in the revised version.

Reviewer #2 (Remarks to the Author):

This manuscript described a hollow fiber/gel-hybrid system for glucose-responsive insulin delivery. The authors systematically studied the formulation of the glucose-responsive gel to design a temperature-independent insulin delivery system. This system was of inherent stability, diffusion-dependent scalability, week-long sustainability and acute glucose-responsiveness. Some revisions were needed before it was considered for publication.

1. NIPMAAm was described as the thermos-sensitive monomer. However, under the formulation of l:m:n:o = 39.56:16.95:38.48:5.00 in Fig. 2 (B), the gel was slightly thermos-sensitive at the glucose concentration of 0~500 mg/dL. Could the authors explain the contribution of NIPMAAm in this formulation?

2. A table for the mole ratios of all the monomers was needed to further describe the pie charts in Fig. 2 and SI Fig. 1.

3. The quality of some figures should be improved. For example, the numbers and characters on the axes of SI Fig. 3 were not clear.

4. The sentence "To a previously reported copolymer system...formation with FPBA" (Line 114-117) was not proper. The authors should rewrite it and give appropriate citations.

Reviewer #3 (Remarks to the Author):

Looks like an interesting paper, but it is lacking in background information on hydrogel chemistry. why use a temperature-responsive monomer like NIPAM? why use a fluorinated phenylboronic acid containing monomer? How were the comonomer mole ratio chosen? does the hydrogel swell or shrink with increase in glucose concentration?

Reviewer 1:

We would like to thank you for your helpful and instructive comments. In light of your comments, we have revised the manuscript as follows. All changes made in the text have been highlighted in yellow color.

General comment:

This manuscript by Akira Matsumoto and colleagues is extremely well-done, thorough, and highlights a previously unappreciated role for electronics-free insulin delivery system. It is novel and with some modest revisions is likely to be of interest to a broad readership.

Reviewer's comment 1:

Other problems:

Fig.1 should be either moved to Supplemental or results, currently, it is in the introduction.

The introduction section is too lengthy. My advice is to make it short in the revised version.

Our response:

To shorten the Introduction section, we moved Fig. 1 to Results section along with the explanatory sentences in the revised version. Thanks again for your advice.

Reviewer 2:

We would like to thank you for your helpful and instructive comments. In light of your comments, we have revised the manuscript as follows. All changes made in the text have been highlighted in yellow color.

General comment:

This manuscript described a hollow fiber/gel-hybrid system for glucose-responsive insulin delivery. The authors systemically studied the formulation of the glucose-responsive gel to design a temperature-independent insulin delivery system. This system was of inherent stability, diffusion-dependent scalability, week-long sustainability and acute glucose-responsiveness. Some revisions were needed before it was considered for publication.

Reviewer's comment 1:

NIPMAAm was described as the thermos-sensitive monomer. However, under the formulation of l:m:n:o = 39.56:16.95:38.48:5.00 in Fig. 2 (B), the gel was slightly thermos-sensitive at the glucose concentration of 0~500 mg/dL. Could the authors explain the contribution of NIPMAAm in this formulation?

Our response:

Thank you for raising this point. The main-chain poly(NIPMAAm) without any comonomers has its lower critical solution temperature (LCST) at round 40 °C. However, as mentioned in Line 124-126 along with references, once copolymerized with boronate components, the apparent LCST decreases due to the hydrophobicity of the boronate component. This is what is observed in Fig. 2 (B) (l:m:n:o = 39.56:16.95:38.48:5.00). Even so, the temperature-dependency for this formulation becomes apparent when observed at lower range (below 25 °C). Inspired by your comment, the following explanatory sentences were added to the caption of SI Fig. 1 in the revised version.

Line 636-643

In each formulation, the gel tends to swell (become more hydrated) with increase of glucose and decrease of temperature. It is also observed that the relative volume of the gel increases (network-loosening) with increased content of NHEAAm, i.e., e → f → g → h or i → j → k → l, and decreases with increased content of FPBA, i.e., or d → f → i → m. With these relationships in mid, our screening criteria was to find a formulation that can (1) minimize the temperature-dependency for the glucose range of 100-500 mg/dL, while also (2) minimizing the network-loosening for lower range of glucose in order to avoid unwanted leakage of insulin.

Reviewer's comment 2:

A table for the mole ratios of all the monomers was needed to further describe the pie charts in Fig. 2 and SI Fig. 1.

Our response:

We agree with you on this point. The table was made and provided as SI Table 1 in the revised version.

Reviewer's comment 3:

The quality of some figures should be improved. For example, the numbers and characters on the axes of SI Fig. 3 were not clear.

Our response:

We appreciate this comment. This Figure has been replaced by a better-quality version.

Reviewer's comment 4:

The sentence "To a previously reported copolymer system...formation with FPBA" (Line 114-117) was not proper. The authors should rewrite it and give appropriate citations.

Our response:

That sentence has been modified to say, "To a previously reported copolymer system [14], graded amounts of *N*-hydroxyethylacrylamide (NHEAAm) was introduced, which is a hydrophilic comonomer bearing hydroxyl group to serve as an intermolecular cross-linker *via* boronate ester formation with FPBA (Fig. 2A)". Also, the new sentence is accompanied by a reference [14].

Reviewer 3:

We would like to thank you for your helpful and instructive comments. In light of your comments, we have revised the manuscript as follows. All changes made in the text have been highlighted in yellow color.

General comment:

Looks like an interesting paper, but it is lacking in background information on hydrogel chemistry.

Reviewer's comment 1:

Why use a temperature-responsive monomer like NIPAM?

Our response:

The use of a thermos-sensitive main-chain component is necessary in order to achieve a sharp transition in hydration, i.e., skin layer formation. This has been mentioned in the revised version (Line 114-116).

Reviewer's comment 2:

Why use a fluorinated phenylboronic acid containing monomer?

Our response:

We added a sentence (Line 110-112) to explain why we chose such a structure, that is, "one key aspect in the optimized chemical structure is the use of a phenylboronic acid (PBA) derivative possessing para-carbamoyl and meta-fluoro substituents (FPBA), resulting in the pKa value of 7.2, which is suitable for the glucose-sensitivity at the physiological pH (7.4)".

Reviewer's comment 3:

How were the comonomer mole ratio chosen?

Our response:

Thank you for raising this important point. The comonomer molar ratio was chosen based on the following two specific criteria;

1. a minimized temperature-dependency especially for the glucose range of 100-500 mg/dL,
2. a minimized network-loosening at lower range of glucose (note that the relative volume of the gel increases with increased content of NHEAAm, which could lead to unwanted leakage of insulin).

In spired by this comment, these criteria have been described in the caption of SI Fig. 1 in the revised version as follows;

Line 636-643

In each formulation, the gel tends to swell (become more hydrated) with increase of glucose and decrease of temperature. It is also observed that the relative volume of the gel increases (network-loosening) with increased content of NHEAAm, i.e., $e \rightarrow f \rightarrow g \rightarrow h$ or $i \rightarrow j \rightarrow k \rightarrow l$, and decreases with increased content of FPBA, i.e., or $d \rightarrow f \rightarrow i \rightarrow m$. With these relationships in mind, our screening criteria was to find a formulation that can (1) minimize the temperature-dependency for the glucose range of 100-500 mg/dL, while also (2) minimizing the network-loosening for lower range of glucose in order to avoid unwanted leakage of insulin.

Reviewer's comment 4:

Does the hydrogel swell or shrink with increase in glucose concentration?

Our response:

As demonstrated in Fig. 2b, the hydrogel tends to swell with increase in glucose concentration. We believe the above description (Line 636-643) made this point clear.

Reviewers' comments:

Reviewer #1 (Remarks to the Author):

This manuscript described a hollow fiber/gel-hybrid system for glucose-responsive insulin delivery. The authors systemically studied the formulation of the glucose-responsive gel to design a temperature-independent insulin delivery system. This system was of inherent stability, diffusion-dependent scalability, week-long sustainability and acute glucose-responsiveness. All the comments were replied, but still some mistakes existed in the manuscript. Thus, the further revision was needed before this manuscript was considered for publication.

1. The following questions needed the authors' explanation.

1.1 The concept of "skin layer" (Line 256) was introduced to explain the phenomenon of glucose-dependent insulin delivery in this system. Could the authors provide proofs (such as SEM images) for the formation of the skin layer in this system?

1.2 What's the relationship between the "skin layer" and SI Fig. 1?

1.3 In SI Fig. 1, most of the gels slightly swelled (or even shrank) when the glucose concentrations increased from 100 to 500 mg/dL. Could the authors explain the relationship between SI Fig. 1 and the mechanism of glucose-dependent insulin delivery?

2. In Fig. 2(a), the structural formula of NHEAAm was wrong. -CH₃ should be deleted.

3. All the No.s of the figures and tables should be carefully checked. For example, "Fig. 3B" should be "Fig. 3C" in Line 387-405.

4. Writing should be improved. For example,

4.1 In SI Tab. 1, the sum of the molar fractions in each column should be 100%; "NHEAA" should be "NHEAAm".

4.2 The contents in Line 506-515 and Line 640-650 were almost the same. The authors should delete the unnecessary part.

Reviewer 1:

Thank you for your instructive comments. We have taken all the comments into consideration and revised manuscript accordingly. All changes made in the text have been highlighted in yellow color.

General comment:

This manuscript described a hollow fiber/gel-hybrid system for glucose-responsive insulin delivery. The authors systemically studied the formulation of the glucose-responsive gel to design a temperature-independent insulin delivery system. This system was of inherent stability, diffusion-dependent scalability, week-long sustainability and acute glucose-responsiveness. All the comments were replied, but still some mistakes existed in the manuscript. Thus, the further revision was needed before this manuscript was considered for publication.

Reviewer's comment 1:

1. The following questions needed the authors' explanation.

1.1 The concept of "skin layer" (Line 256) was introduced to explain the phenomenon of glucose-dependent insulin delivery in this system. Could the authors provide proofs (such as SEM images) for the formation of the skin layer in this system?

Our response:

The formation of skin layer is a nonequilibrium dehydration process, allowed to occur in a limited window of physicochemical environment. Therefore, it is not possible to observe it on dried state (such as SEM imaging). When the gel is prepared in a macroscopic dimension, the skin layer formation can be appreciated by naked eyes as a change in optical appearance due to light-scattering; please check the attachment Reviewer's Only Fig. #1 from reference [15] (the gel formulation is identical to (a) in SI Fig. 1 in the present manuscript). Another paper we have previously published (ref. [14]) has provided characterization of "skin layer" in detail. Please find the attachment Reviewer's Only Fig. #2 (the gel formulation in this Figure is identical to (a) in SI Fig. 1 in the present manuscript). In this experiment, a reporter fluorescence dye 8-anilino-1-naphthalene sulfonic acid (ANS) was utilized for its ability to increase the intensity when exposed to the environment of decreased local polarity (distinctive of dehydration). By confocal microscopy observation, we could reveal the dynamics and detailed geometry of the skin layer. In fact, we have carried out the same experiment using a hollow fiber/gel-hybrid system. However, this experiment turned out difficult because the neighboring, permanently hydrophobic fiber matrix outweighs the change in ANS intensity from the gel.

Reviewer's comment 2:

1.2 What's the relationship between the "skin layer" and SI Fig. 1?

Our response:

As requirement for the skin layer formation, “network loosening” at lower glucose (up to 100 mg/dL) must be prevented. In other words, if the degree of hydration (Y axis in phase diagrams) at lower glucose is not sufficiently low, then the gel is no longer able to form the skin layer (Chem. Lett., 2016, 45(4), 460-462. DOI: [10.1246/cl.151177](https://doi.org/10.1246/cl.151177)). In this regard, SI Fig. 1 provides useful information. It is also important to point out that the phase diagrams are equilibrium state information, whereas the skin layer is a nonequilibrium event. Even so, from SI Fig. 1 we can estimate whether or not each formulation meets the requirement for the skin layer formation, by looking at Y axis values at low glucose conditions relative to that for the formulation (a). To make this point clear, we added the following sentence to the text (Line 133-134).

Careful attention was paid to the retaining of sufficient dehydration level (Y axis) at lower glucose conditions so that the gel would not compromise on the ability of skin layer formation.

Reviewer’s comment 3:

1.3 In SI Fig. 1, most of the gels slightly swelled (or even shrank) when the glucose concentrations increased from 100 to 500 mg/dL. Could the authors explain the relationship between SI Fig. 1 and the mechanism of glucose-dependent insulin delivery?

Our response:

In the gel science community, the most common theory to describe solute diffusion property in gels is “free-volume” theory. That is, under premise of constant polymer chain mobility and the polymer-solute interaction, the apparent diffusion coefficient of a solute (D_{app}) within hydrogels is a function of the size of the solute relative to that of the mesh size (or free-volume). As a result, a general observation is that D_{app} decreases as the swelling degree of the gel decreases. Indeed, the inverse of the swelling degree governs an exponential decrease of D_{app} . This means that, in an optimally free-volume-available network (which also depends on the molecular weight of the solute), one can control D_{app} exponentially and thus even a slight change in the swelling degree can dramatically affect the solute diffusion property. Here, analysis of phase diagrams on SI Fig. 1 (comparing Y axis values at low glucose conditions relative to those for the formulation (a)) helps to choose a right gel formulation. To make this point clear, along with the above sentence (Line 133-134), additional explanation is provided in caption of SI Fig. 1.

2. In Fig. 2(a), the structural formula of NHEAAm was wrong. $-CH_3$ should be deleted.

Our response:

We do appreciate your careful check. The monomer structure has been corrected.

3. All the No.s of the figures and tables should be carefully checked. For example, “Fig. 3B” should be “Fig. 3C” in Line 387-405.

Our response:

These typos have been corrected. Thanks for pointing it out.

4. Writing should be improved. For example,

4.1 In SI Tab. 1, the sum of the molar fractions in each column should be 100%; “NHEAA” should be “NHEAAm”.

Our response:

This has been caused because all figures have been rounded off to the second decimal place.

This is mentioned in the table caption of the revised version. Also, the miswriting of the monomer abbreviation has been corrected.

4.2 The contents in Line 506-515 and Line 640-650 were almost the same. The authors should delete the unnecessary part.

Our response:

Line 640-650 has been deleted.

A

Reviewer's Only Fig. 2 (from reference [15]). Images of a smart gel formed in a macroscopic slab shape equilibrated under hyperglycemic (1000 mg/dl) (left) and no-glucose (right) conditions.

Reviewer's Only Fig. 1 (from reference [14]). (a) Time-course (*left*) transmittance and (*right*) 8-anilino-1-naphthalene sulfonic acid (ANS) fluorescence top-view images of a cylinder-shaped piece of gel when changing the glucose concentration under pH 7.4 and 37 °C. Indicated in each transmittance image are elapsed times after initial lowering of the concentration from 2 g L⁻¹ to 1 g L⁻¹. In fluorescence images, appearance of the skin layer can be seen as increased blue colour intensity of ANS correlating to the dehydration on the gel surface. Conversely, regression in the intensity upon re-increase of the glucose (t = 90 min.) is indicative of disappearance of the skin layer. (b) Changes in the ANS intensity profiles at 150 μm depth cross-section (from the gel surface) during the growth of the skin layer. (c) Schematic representation of the cross-sectional density distribution of the polymer network in the gel when reversibly forming the skin layer.